# Identification and Expression Analysis of *SLAC*/*SLAH* Gene Family in *Brassica napus* L.

**DOI:** 10.3390/ijms22094671

**Published:** 2021-04-28

**Authors:** Yunyou Nan, Yuyu Xie, Ayub Atif, Xiaojun Wang, Yanfeng Zhang, Hui Tian, Yajun Gao

**Affiliations:** 1College of Natural Resource and Environment, Northwest A&F University, Yangling 712100, China; nanyunyou@nwafu.edu.cn (Y.N.); xieyuyuC@163.com (Y.X.); Atifayub.gcuf@gmail.com (A.A.); xiaojunwang@nwafu.edu.cn (X.W.); 2Hybrid Rapeseed Research Center of Shaanxi Province, Yangling 712100, China; zhangyfcl@163.com; 3Key Laboratory of Plant Nutrition and the Agri-Environment in Northwest China, Ministry of Agriculture, Yangling 712100, China

**Keywords:** *Brassica napus*, SLAC/SLAH, nitrate, abiotic stress, expression analysis

## Abstract

Slow type anion channels (SLAC/SLAHs) play important roles during anion transport, growth and development, abiotic stress responses and hormone responses in plants. However, there is few report on SLAC/SLAHs in rapeseed (*Brassica napus*). Genome-wide identification and expression analysis of *SLAC*/*SLAH* gene family members were performed in *B. napus*. A total of 23 *SLAC*/*SLAH* genes were identified in *B. napus*. Based on the structural characteristics and phylogenetic analysis of these members, the SLAC/SLAHs could be classified into three main groups. Transcriptome data demonstrated that *BnSLAH3* genes were detected in various tissues of the rapeseed and could be up-regulated by low nitrate treatment in roots. BnSLAC/SLAHs were exclusively localized on the plasma membrane in transient expression of tobacco leaves. These results will increase our understanding of the evolution and expression of the SLAC/SLAHs and provide evidence for further research of biological functions of candidates in *B. napus*.

## 1. Introduction

Slow and rapid types of anion channels have been found in plants, which are expressed in guard cells and mediate slow and fast anion currents [1]. Slow type anion channel proteins (SLAC/SLAH) are one of the subfamilies of nitrate transporters. There are five SLAC/SLAH members in *Arabidopsis* (*Arabidopsis thaliana*) and they play an important role in stress signaling, growth and development, and hormonal response [2]. The interaction of AtSLAC1 and AtSLAH3 with different kinase phosphatases is associated with water stress signals [3,4,5]. In addition, SLAC1 is predominantly distributed in guard cells and is phosphorylated by Open stomata 1 (OST1) kinase, resulting in anion efflux from guard cells mediating stomatal closure [6,7]. In terms of hormonal signaling, abscisic acid (ABA) activates SLAC1 from direct interactions with OST1 and Calcium-dependent protein kinases (CPKs), and contributes to increase drought tolerance by regulating stomatal closure [5]. SLAC/SLAHs are also involved in the process of plant growth and development. For instance, SLAH3 phosphorylated by calcium-dependent protein kinase such as CPK2 and CPK20 in Arabidopsis regulated pollen tube growth [8]. In Arabidopsis, *SLAC1* is mainly expressed in guard cells but weakly expressed in root/stamen/young silique, whereas *SLAH3* is weakly expressed in guard cells and stamens, but strongly expressed in root [9]. SLAH2, the closest homologous protein to SLAH3, exclusively absorbs nitrate which is different from other SLAC/SLAH members with both absorption to nitrate and chloride [10]. *SLAH1* is expressed in root, hypocotyl and stamen, and controls Cl^−^ transport from root to shoot without affecting the absorption of NO_3_^−^, which makes *SLAH1* a promising candidate gene for salt-tolerant plant biotechnology breeding [11,12]. *SLAH4* is strongly expressed near the root tip, [13]. SLAC1 and SLAH3 have permeability ratios of 10 and 20 to NO_3_^−^/Cl^−^, respectively, when they are expressed in *Xenopus* oocytes [13]. SLAH3 shows a higher preference for nitrate than SLAC1, which is supposed to be a nitrate channel protein and functions in nitrate-dependent alleviation of ammonium toxicity [9]. The latest studies showed that sucrose non-fermenting 1-related protein kinase 1.1 (SnRK1.1) phosphorylated the C-terminal of SLAH3 and negatively regulated the process of SLAH3 involved in nitrate-dependent ammonium toxicity alleviation [14].

In rice (*Oryza sativa*), a total of 9 SLAC/SLAH members have been identified. Protein kinase OsSAPK8 is an important activator of OsSLAC1, and OsSLAC1 has nitrate selectivity [15]. In barley (*Hordeum vulgare*), HvSLAC1 requires nitrate to achieve ABA-induced stomatal closure. HvSLAC1 and other slow type anion channels in monocot have nitrate-dependent gating characteristics after the separation of monocot and dicot by the evolution of TMD3 series motifs in the trans-membrane domain [16]. ZmSLAC1 is a nitrate selective anion channel without obvious permeability to chloride, sulfate and malatein maize [17]. In pear (*Pyrus bretschneideri*), PbrSLAH3 are localized in the plasma membrane without expression in flower. Moreover, PbrSLAH3 has a strong selective absorption to nitrate and no permeability to chlorine, which is similar to ZmSLAC1. PbrSLAH3 regulated by PbrCPK32 is involved in the absorption and transport of nitrate in pear roots [18]. PttSLAH3 of poplar is not activated by protein kinase phosphorylation to absorb nitrate and chloride ions [19]. These studies indicate that SLAC/SLAHs has the nitrate selectivity and permeability and can regulate stomatal closure by regulating anion efflux from guard cells in response to ABA signals.

At present, more and more studies have focused on the slow anion channel proteins, especially those involved in the absorption and transport of nitrate. SLAC/SLAH family members have been identified and studied in several species such as *Arabidopsis* [9], rice [15], maize [17], barley [16], tobacco [20], poplar [19], and pear [21]. However, *Brassica napus* (*B. napus*), an important oil crop, has not been reported on SLAC/SLAHs. *B. napus* has a higher nitrogen requirement than other crops [22]. It was reported earlier that *SLAC*/*SLAH*s were highly expressed in roots and had the nitrate-permeable channel activity [9], we thus hypothesized that SLAC/SLAH family members in *B. napus* might be involved in nitrate uptake and transport. Therefore, the main objective of this study is to identify the *SLAC*/*SLAH* gene family of *B. napus* and analyze the structure, physicochemical properties, evolutionary relationships and expression profiles of corresponding genes through bioinformatics analysis. It is hoped to provide reference for further research on the biological functions of key candidate genes in nitrogen uptake and utilization of rapeseed.

## 2. Results

### 2.1. Identification and Classification of SLAC/SLAH Genes in B. napus

To identify the members of SLAC/SLAH family in *B. napus*, local protein BLAST using SLAC/SLAH protein sequences from *Arabidopsis* and Hidden Markov Model (HMM) search with conserved model (SLAC1.hmm) as query were performed. A total of 23 genes was identified in *B. napus* after several sequences redundant or without conserved domain (TDT_SLAC1_like) were removed (Table 1). Meanwhile, eleven and 12 SLAC/SLAH family members were identified in *Brassica rapa* (*B. rapa*) and *Brassica oleracea* (*B. oleracea*), respectively (Appendix A). The SLAC/SLAHs in *B. napus* were named according to the order of closest orthologues with *Arabidopsis*. *B. napus SLAC*/*SLAH* genes (hereafter termed as BnSLAC/BnSLAH) mainly included 2 to 6 exons encoding proteins of 329 to 640 residues and about 37.57 to 72.59 kDa molecular weight and 6 to 9.7 Isoelectric point (pI) value. These proteins were predicted to localize in plasma-membrane with 6 to 10 putative transmembrane domains (TMDs).

### 2.2. Phylogenetic Analysis of the SLAC/SLAHs between B. napus and Other Species and Identification of Evolutionary Selection Pressure on BnSLAC/SLAHs

To classify the SLAC/SLAHs and investigate their evolutionary relationships identified in *B. napus*, *B. rapa* and *B. oleracea*, phylogenetic analysis were performed to infer a functional relationship among them. A phylogenetic tree based on SLAC/SLAH protein sequences of *A. thaliana*, *O. sativa*, *B. napus*, *B. rape* and *B. oleracea* was constructed by the Neighbor-joining (NJ) method, which showed that the SLAC/SLAHs from the 5 species could be divided into three main groups, including SLAC, SLAH2/3 and SLAH1/4 (Figure 1). In phylogeny, group SLAH2/3 formed the largest cluster with 11 members in *B. napus*, 6 members in *B. rapa*, 5 members in *B. oleracea*, 2 members in *A. thaliana* and 6 members in *O. sativa*.

To characterize the selection pressure on the BnSLAC/SLAHs during the evolutionary process, we used the orthologous *SLAC*/*SLAH* gene pairs between *B. napus* and *A. thaliana* to determine the values of synonymous (Ks) and non-synonymous (Ka) nucleotide substitution rates, and Ka/Ks (Table 2). The Ka values of BnSLAC/SLAHs ranged from 0.0567 (BnSLAH2-1) to 0.2183 (BnSLAH4-5 and BnSLAH4-6) with an average of 0.1063, and the Ks values of BnSLAC/SLAHs ranged from 0.0677 (BnSLAH4-1) to 0.4910 (BnSLAH3-6) with an average of 0.3518. Further, we found that the Ka/Ks values of most BnSLAC/SLAHs were less than 1.0 except that of BnSLAH4 subfamily members were more than 1.0. Therefore, we presumed that the BnSLAC1s, BnSLAH1s, BnSLAH2s, BnSLAH3s might have experienced a very strong purify selection to preserve their function while BnSLAH4s were positively selected.

### 2.3. Chromosomal Location and Duplication Patterns

*B. napus* (AACC, 2n = 38) is an allotetraploid hybrid species of *B. rapa* (AA, 2n = 20) and *B. oleracea* (CC, 2n = 18) [23]. All the *BnSLAC*/*SLAH* genes were unevenly distributed on the rest 13 of 19 chromosomes according to the position of the start and stop codons (Figure 2) except for chromosomes A03-A05, A8, A10 and C06. Among them, 9 *BnSLAC*/*SLAH* genes were mapped to the chromosomes of A genomes while 14 members were found in the C genomes. Chromosome A09 and C01 included a maximum of 3 *BnSLAC*/*SLAH* genes, respectively. However, Chromosome A02, A07, C02, C03 and C05 only had a single BnSLAC/SLAH member. Genomic duplications in plants could be either whole-genome duplications (WGD) as large scale or tandem and segmental duplications as small scale. Tandem duplications can form on the same chromosome while segmental duplications events often occur on the different chromosomes. The number of *BnSLAC*/*SLAH*s in *B. napu*s is different from that in *B. rapa* and in *B. oleracea* genomes, indicating that the slow type anion channels in *B. napus* underwent expansion and evolution in comparison with its ancestor species, *B. rapa* and *B. oleracea*. This study identified several tandem and segmental duplications between some *BnSLAC*/*SLAH* gene pairs. *BnSLAH4-5* and *BnSLAH4-6* gene pair located on the chromosome C01 generated from tandem duplications was found whereas 2 *BnSLAC1s* (*BnSLAC1-1* and *BnSLAC1-2*), 2 *BnSLAH1*s (*BnSLAH1-2* and *BnSLAH1-3*), 2 *BnSLAH2*s (*BnSLAH2-1* and *BnSLAH2-2*), 5 *BnSLAH3*s (*BnSLAH3-2* to *BnSLAH3-6*) and 4 *BnSLAH4*s (*BnSLAH4-1* to *BnSLAH4-4*) were identified to be segmental duplicated genes (Figure 2). These results indicated that segmental duplication events functioned as a major driving force for the expansion of *SLAC*/*SLAH*s in *B. napus*.

### 2.4. Gene Structure and Conserved Motif Analysis of the SLAC/SLAH Family Members in B. napus

The phylogenetic tree of BnSLAC/SLAH protein sequences illustrated the evolutionary relationships of these members in *B. napus* (Figure 3). All the coding sequences of the *BnSLAC*/*SLAH* genes were spaced by introns with numbers varying from 1 to 5. The number of introns and the lengths of individual exons had the high similarity within the same subfamily, except that *BnSLAH4* genes had different numbers of exons ranging from 2 to 5 (Figure 3 middle). Motif analysis was performed for the most conserved 10 motifs sequences in 23 BnSLAC/SLAH proteins using MEME tool. Most of the closely related members in the phylogenetic tree shared common motif compositions (Figure 3 right). BnSLAH3 subfamily contained motif 1 to motif 10 while BnSLAH2 mainly had motif 1-motif 5 and motif 7. Motif 1 to motif 7 were widely distributed in the BnSLAH1, BnSLAH4 and BnSLAC1 subfamilies. These conserved motifs are considered to possibly have functional and/or structural roles in active proteins.

### 2.5. Promoter cis-Acting Element Analysis and Interaction Protein Prediction

Attempting to understand the transcriptional regulation mechanisms of the identified *BnSLAC*/*SLAH* genes, it was essential to have insights about the upstream promoter region (Ravel et al. 2015). Two-kb upstream region from the initiation codon of *BnSLAC*/*SLAH* genes was obtained and supplied to the Plant CARE database to investigate of cis-regulatory elements. A total of 42 different cis-elements associated with light responsive, stress responsive, the phyto-hormone responsive and growth regulation have been identified in upstream regions of 23 *BnSLAC*/*SLAH* genes. A heatmap was constructed based on presence or absence of regulatory elements in each corresponding gene to better characterize the large number of motifs (Figure 4). These results indicated that complex regulatory networks may be implicated in the transcriptional regulation of BnSLAC/SLAHs. Cis-regulatory elements, CAAT-box and TATA-box, were commonly shared by all *BnSLAC*/*SLAH* genes. Obviously, CAAT- and TATA boxes are two common cis-regulatory elements in upstream regions of eukaryotic genes. Box4, G-box and TCT-motif elements responding to light existed in the 2-kb upstream region of more than approximately 70% *BnSLAC*/*SLAH* genes. Most *BnSLAC*/*SLAH* genes contained ABRE element (ABA responsive) except for *BnSLAH2-1*, *BnSLAH2-3* and *BnSLAH3-2*. *BnSLAH2* and *BnSLAH3* subfamily genes except for *BnSLAH2-1* contains TGACG-motif element involved in MeJA response. Moreover, eleven *BnSLAC*/*SLAH*s harbored drought responsive cis-elements (MBS) while 8 *BnSLAC*/*SLAH*s contained low-temperature responsive cis-element (LTR).

To further identify the proteins potentially interacting with the SLAC/SLAH family members, we constructed a protein interaction network involving direct (physical) and indirect (function) association by using the STRING database based on either known experimental or predicted interactions. All the SLAC/SLAH proteins consistently interacted with OST1. Besides, CPK6, CPK 21 and CPK23 were also observed to interact with the SLAC/SLAH proteins (Appendix A). Interestingly, BnSLAH1 and BnSLAH3 were predicted to interact with nitrate excretion transporter 1 (NAXT1) which was one of the nitrate transporters.

### 2.6. Synteny Analysis

In the present study, we also identified 11 and 12 *SLAC*/*SLAH* genes in the *B. rapa* and *B. oleracea* genomes, respectively. To further infer the phylogenetic mechanisms of the BnSLAC/SLAHs, we constructed a comparative syntenic map of *B. napus* and its ancestors (*Arabidopsis*, *B. rapa* and *B. oleracea*). Collinearity analysis revealed that there were strong orthologs of *SLAC*/*SLAH* genes between *B. napus* and the other three ancestral species (Figure 5). Nine and 4 of the *SLAC*/*SLAH* genes in the A subgenome of *B. napus* showed syntenic relationships with 9 and 4 *SLAC*/*SLAH* genes in the *B. rapa* and *Arabidopsis* genomes, respectively. In contrast, eight and 2 of the genes in the *B. napus* C subgenome were syntenic with 9 and 2 of the *B. oleracea* and *Arabidopsis* genomes, respectively. Additionally, 10 pairs of *BnSLAC*/*SLAH* genes are paralogs to each other in *B. napus*. The fact that nearly all of the homologous *BrSLAC*/*SLAH*s and *BoSLAC*/*SLAH*s maintained a syntenic relationship with *BnSLAC*/*SLAH*s suggested that whole-genome duplication (polyploidy) also played a major driving force for *BnSLAC*/*SLAH*s evolution besides segmental duplication.

### 2.7. Subcellular Localization of the SLAC/SLAH Genes of B. napus

All of the BnSLAC/SLAHs were predicted to localize on the plasma membrane. To confirm the above prediction, three *BnSLAC/SLAH* genes were cloned and fused to the N-terminus with the GFP protein driven by the CaMV 35S promoter. The fusion proteins (BnSLAC/SLAH-GFP) and control (empty vector) were transiently transformed into tobacco leaves. Microscopic visualization showed that green fluorescence was distributed throughout the whole cell when the empty vector was used. The green fluorescence was exclusively detected on the plasma membrane by confocal microscopic when the vectors contained BnSLAH1-1, BnSLAH3-2 and BnSLAH3–3 while that was detected both on the plasma membrane and nucleus in the control (Figure 6). Thus, the BnSLAC/SLAHs might have a similar subcellular localization pattern that they were all membrane protein.

### 2.8. Expression Level of the SLAC/SLAH Genes in B. napus

The expression level of genes usually affects their biological function. In order to better investigate the functions of the *BnSLAC/SLAH* genes, we analyzed the expression patterns of the *BnSLAC/SLAH* genes in 12 different tissues based on the RNA-seq data from the *B. napus* cv. ‘Zhongshuang 11′ (BioProject ID PRJNA394926) (Figure 7, Appendix A). According to the results, several genes belong to the *BnSLAH3*, *BnSLAH1* and *BnSLAC1* subfamilies had a higher expression in most tissues than others, which shared similar expression patterns in specific tissue. Especially, *BnSLAH3-2*, *BnSLAH3-3* and *BnSLAH3-4* were highly expressed in the pistil, root, stamen, silique and leaf. The transcripts of the *BnSLAH4* and *BnSLAH2* subfamilies genes were weakly expressed in the 12 different tissues. Moreover, *BnSLAH1-1* was specifically expressed in the stamen, which was similar to *BnSLAH1-2*.

To investigate the mechanisms of nitrate transport and uptake of slow anion channel in *B. napus*, we selected several *BnSLAC/SLAH* genes for further study. The expression of *BnSLAC/SLAH* genes in leaves and roots of rapeseed seedlings under high nitrate (HN 7.5 mM) and low nitrate (LN 0.19 mM) concentrations was determined as was shown in Figure 8. In leaves, the expression level of the *BnSLAC1-1* and *BnSLAH3-2* increased about 10 and 5 folds after 7 d LN treatment. In addition, the expression level of *BnSLAH1-1* and *BnSLAH3-3* evaluated about 2 to 4 folds after 3 h and 12 h LN treatment while that of *BnSLAH3-4* had no obvious change. In roots, the expression of *BnSLAC1-1* was increased 3.5 times at 24 h after LN treatment but not significantly changed at other treatment time points. The expression levels of *BnSLAH1-1* were significantly up-regulated especially at 3 h after low nitrogen treatment. In addition, *BnSLAH3-2* was significantly up-regulated 2 to 3 times under 12 h and 24 h low N treatment. Besides, 3 h and 12 h LN treatment induced a 2 to 4-fold up-regulation of *BnSLAH3-3* and *BnSLAH3-4* expression. Therefore, the slow anion channel proteins are able to respond to low nitrogen signal in a short time, which may be involved in nitrate uptake and transport in *B. napus*.

ABA is a phyto-hormone involved in a variety of abiotic stress response. By analyzing the Cis-acting elements of the *BnSLAC/SLAH* promoter, we found that most members contained ABA response elements (ABRE). Therefore, we detected the change of gene expression in rapeseed seedlings treated with 100 uM ABA (Figure 9). The results showed that the expressions of *BnSLAC1-1* and *BnSLAH3-4* genes were significantly decreased after 12 h ABA treatment in leaves. *BnSLAH3-2* was up-regulated 3–4 folds at 3 h and 24 h after ABA treatment, while the expression of *BnSLAH3-3* displayed 58-fold up-regulation at 12 h after ABA treatment in leaves. In roots, *BnSLAC1-1* was up-regulated about 2.5 folds after 24 h ABA treatment, while *BnSLAH1-1*, *BnSLAH3-2*, *BnSLAH3-3* and *BnSLAH3-4* showed down-regulation trend after 3 h, 12 h and 24 h ABA treatment.

## 3. Discussion

In this study, 23, 11 and 12 SLAC/SLAH family members were identified in *B. napus*, *B. rapa* and *B. oleracea* respectively using 5 AtSLAC/SLAH sequences as queries. *B. napus* is a heterotetraploid formed by natural multiplication and interspecific hybridization between *B. rapa* and *B. oleracea* [23]. However, 9 *BnSLAC/SLAH* members are located in the A sub-genome of *B. napus*, which is less than the 11 *BrSLAC/SLAH* members identified in *B. rapa*. In addition, 14 *BnSLAC/SLAH* members are located in the C sub-genome of *B. napus*, which is more than 12 *BoSLAC/SLAH* members of *B. oleracea*. This may be due to the occurrence of genome-wide duplication events, tandem and segmental duplication events, resulting in partial gene deletions and increases during evolution. To explore the evolutionary process of BnSLAC/SLAHs, we investigated phylogeny, gene duplication events and syntenic relationships between *B. napus* and other species. These SLAC/SLAHs of *Arabidopsis*, rice, *B. napus*, *B. rapa*, and *B. oleracea* were divided into three groups, including SLAC1, SLAH2/3 and SLAH1/4, which was consistent with the classification of SLAC/SLAHs in *Arabidopsis* [9]. We found that expansion of *SLAC/SLAH* genes was attributed to WGD and segmental duplication event, and Ka/Ks analysis indicated that all the SLAC/SLAHs except BnSLAH4s were purified to evolve. However, previous study indicated that SLAC/SLAH developed along with the stoma evolution [24]. These results implies that SLAC/SLAH plays a crucial role in regulating stomatal movement. We found that BnSLAH3 members contain 5 exons and 10 conserved motifs, while BnSLAH1 genes only have 2 exons, which indicate BnSLAH3 may have functional diversity during growth and development in plants. *SLAC/SLAH* genes have a wide range of functions in plant growth and development. In addition, the result of promoter analysis showed that 2-kb upstream region of *BnSLAC/SLAH* genes contained different kinds of cis-elements involved in the light response, phyto-hormone response (including ABA, IAA, GA, MeJA and SA), drought response, low temperature response and growth regulation. *BnSLAC/SLAH* may function in the abiotic stress and growth regulation. *AtSLAC1* encodes a plasma membrane-localized protein that is highly permeable to malate and chloride in *Arabidopsis* [25]. In this study, several *BnSLAC/SLAH* genes were selected for subcellular experiment. The observation of confocal microscopy suggested that BnSLAH1-1, BnSLAH3-2 and BnSLAH3-3 were localized on the plasma membrane, which is consistent with the results observed in *Arabidopsis* and pear as well as the prediction of subcellular localization in *B. napus*.

Five slow anion channel proteins, SLAC1 and its homologues SLAH1-SLAH4, were found in the model plant *Arabidopsis*, which were differentially expressed in leaves, stamens and roots [9]. RNA-seq results in *B. napus* showed that *BnSLAH3-2*, *BnSLAH3-3* and *BnSLAH3-4* genes were highly expressed in stamen, pistil, root, leaf, silique and sepal, while *BnSLAC1-1* and *BnSLAC1-2* genes showed a high expression level in the sepal, which is different from the expression patterns of AtSLAC1. This suggests that several *BnSLAH3* genes may play an important role in the growth and development of rapeseed in comparison with other *BnSLAC/SLAH* members. Previous studies have shown that the plant slow anion channel proteins are involved in regulating the electrical conductivity of various anions such as chloride and nitrate [25], and function in stress signal, growth and development, and hormonal response [2]. Among them, more studies have focused on SLAC1 and SLAH3. AtSLAH3 was reported to be highly expressed in root besides leaves and stamens [9]. The absorption ratio of NO_3_^−^/Cl^−^ of AtSLAH3 was 20 in *Xenopus* oocytes and showed a higher preference for NO_3_^−^, which was consequently considered as a nitrate channel protein [5,6,26]. According to the previous studies and RNA-seq analysis of different tissues in *B. napus*, several *BnSLAH/SLAH* genes were selected for qRT-PCR analysis. The results showed that *BnSLAH3-2* and *BnSLAH3-3* and *BnSLAH3-4* in root could be up-regulated at 12 h after low nitrate treatment (0.19 mM), indicating that the above *BnSLAH3* genes could respond to low nitrate stress in a short time and they might promote nitrate uptake and transport in rapeseed roots. However, There was a report suggesting that high concentration (64 mM) of nitrate induced the expression of SLAC/SLAH genes in pear [21]. It may be attributed to different species, nitrate concentration and treatment time. Accumulation of ABA in wheat induces the expression of NRT2 gene to promote nitrate uptake [27]. Exogenous ABA treatment can improve the yield and nitrogen use efficiency of *B*. *napus* [28]. *BnSLAH3-3* were significantly upregulated about sixty times at 12 h in leaves while *BnSLAH3-2*, *BnSLAH3-3* and *BnSLAH3-3* were down-regulated in roots after ABA treatment. It is suggested that BnSLAH3s in leaves stimulated by exogenous ABA signals may be mainly involved in stomatal movement rather than nitrate uptake in roots.

AtSLAC1 and AtSLAH3 were activated by the calcium-independent protein kinase OST1 as well as by CPKs [29,30,31]. PbrSLAH3 showed a high selectivity for nitrate over chloride and was confirmed to interact with PbrCPK32 through yeast two-hybrid and bimolecular fluorescence complementation assays transport [18]. In addition, the prediction of protein interaction displayed that NAXT1, one of the NRT1 family members, was capable of interacting with SLAH3. Nitrogen is an essential nutrient element for plant. Adequate nitrogen uptake from the external environment is a crucial process for plant growth and development [26]. Rapeseed is the third largest oil crop worldwide [32]. *B.napus*, main cultivar of rapeseed, requires more nitrogen fertilizer with low nitrogen utilization efficiency [33,34]. Based on previous reports on the high expression level of *SLAH3* gene in the root system, the strong permeable to nitrate and the interaction between SLAH3 and NAXT1, it was presumed that BnSLAH3s may be involved in the absorption of nitrate in *B. napus*. In this study, 23 BnSLAC/SLAHs were identified, and the results showed that *BnSLAH3* genes were expressed in multiple tissues and could respond to low nitrate in a short time. Therefore, further study will be focus on the biological function of *BnSLAH3* genes and the regulatory mechanism of nitrate absorption and utilization in *B. napus*.

## 4. Materials and Methods

### 4.1. Identification of SLAC/SLAH Gene Family Members in B. napus

To identify the SLAC/SLAH genes in *B. napus*, various database searches were conducted. The SLAC/SLAH protein sequences of *Arabidopsis*, including AtSLAC1 (AT1G12480), AtSLAH1 (AT1G62280), AtSLAH2 (AT4G27970), AtSLAH3 (AT5G24030) and AtSLAH4 (AT1G62262), were downloaded from the TAIR (http://www.arabidopsis.org/, accessed on 26 January 2021) [35]. In addition, the SLAC/SLAH protein sequences of rice, such as Os04g48530.1, Os01g43460.1, Os05g13320.1, Os01g28840.1, Os01g12680.1, Os07g08350.1, Os05g50770.2, Os01g14520.1 and Os05g18670.1, were downloaded from Phytozome (http://phytozome.jgi.doe.gov/pz/portal.html#, accessed on 26 January 2021). The protein sequences of *B. napus*, *B. rapa* and *B. oleracea* were download from NCBI (https://www.ncbi.nlm.nih.gov/datasets/, accessed on 26 January 2021). The candidate *SLAC/SLAH* genes in *B. napus* were identified by using protein BLAST (https://www.ncbi.nlm.nih.gov/, accessed on 26 January 2021) to search the local Brassica database and using *Arabidopsis* SLAC/SLAH protein sequences as queries and an E-value ≤ 1 × 10^−20^ as a threshold. The HMM profiles of the SLAC1.hmm downloaded from the HMM database (https://www.ebi.ac.uk/Tools/hmmer/, accessed on 26 January 2021) were applied to search for the SLAC1-type sequences in the local protein databases of *B. napus* by performing local HMMER search program with an E-value ≤ 1 × 10^−20^ as a threshold. Furthermore, all the obtained SLAC/SLAH protein sequences were analyzed against the Pfam database (http://pfam.xfam.org/, accessed on 26 January 2021) to verify the presence of SLAC1 domains (PF03595). Protein sequences lacking the SLAC1 domain or having E-values more than 1 × 10^−20^ were removed. The same method was used to identify the SLAC/SLAHs in *B. rapa* and *B. oleracea*.

### 4.2. Phylogenetic and Synteny Analysis of SLAC/SLAH

The phylogenetic trees were constructed based on the SLAC/SLAH protein sequences from *Arabidopsis*, *O. sativa*, *B. rapa*, *B. oleracea* and *B. napus* using the Neighbor-Joining (NJ) method in MEGA6.0 (http://www.megasoftware.net/, accessed on 26 January 2021) [36], and the bootstrap test was carried out with 1000 replicates. DnaSP (DNA Sequence Polymorphism) v5 [37] was used to calculate the ratio of the non-synonymous substitution rate (Ka) to the synonymous substitution rate (Ks) and the Ka/Ks value between paralogous gene pairs.

MCScanX was used to identify syntenic chains according to the homologous pairs as input [38]. The identification of whole-genome (WGD)/segmental, tandem, proximal and dispersed duplications in the SLAC/SLAH gene family were performed by downstream analysis tools in the MCScanX package. The results were displayed using TBtools [39].

### 4.3. Sequence Analysis of SLAC/SLAH Genes/Proteins in B. napus

ProtParam tool was used to calculate the Physicochemical features of SLAC/SLAHs (http://web.expasy.org/protparam/, accessed on 26 January 2021). The structures of the *SLAC/SLAH* genes was analyzed using GSDS server (gsds.cbi.pku.edu.cn/, accessed on 26 January 2021) [40]. Trans-membrane domains of SLAC/SLAHs were predicted by SCAMPI server (http://scampi.cbr.su.se/, accessed on 26 January 2021; Bernsel et al. 2008). The subcellular localization of SLAC/SLAH proteins in *B. napu*s were predicted with the WoLF PSORT online tool (https://www.genscript.com/wolf-psort.html, accessed on 26 January 2021) [41]. Conserved motif of the SLAC/SLAH protein sequences were identified by MEME suite with parameters, where maximum number of motifs was 10, minimum motif width was 6 and maximum motif width was 50 (http://meme-suite.org/, accessed on 26 January 2021) [42]. The phylogenetic tree of SLAC/SLAH in *B. napus* was drawn with the neighbor-joining way by MEGA 6.0 [36].

### 4.4. Chromosomal Location and Duplication of SLAC/SLAH Genes in B. napus

Characterization of the chromosomal location of each *SLAC/SLAH* gene was acquired from NCBI (https://www.ncbi.nlm.nih.gov/datasets/, accessed on 26 January 2021). Based on the position information of *SLAC/SLAH* genes on the chromosomes, all of them were located on the A and C genome chromosomes. *SLAC/SLAH* genes were mapped on the *B. napus* chromosome using Mapchart. Genes located surrounding and parted by less than five genes can be as tandem duplications. The segmental duplications were analyzed by the Maher’s study [43].

### 4.5. Promoter cis-Acting Element Analysis and Interaction Protein Prediction

Two thousand bp upstream regions from the start codon site of *SLAC/SLAH* genes were retrieved from NCBI database and supplied to Plant CARE database for promoter analysis (http://bioinformatics.psb.ugent.be/webtools/plantcare/html/, accessed on 26 January 2021) [44]. STRING (Search Tool for Recurring Instances of Neighboring Genes) v 11.0 (https://string-db.org, accessed on 26 January 2021) web-server was selected to predict and display protein interaction networks of the SLAC/SLAH proteins in *B. napus*.

### 4.6. Subcellular Localization of the SLAC/SLAHs in B. napus

The full-length coding sequences without termination codon of the selected *SLAC/SLAH* genes of *B. napus* were isolated and linked into the pEGFP vector containing GFP reporter (saved in our laboratory). The competent cells of *Escherichia coli* (DH5α) and *Agrobacterium* (LBA4404) were used for the transformation of recombinants. Primers used for gene cloning and vector construction are shown in Appendix A. *Agrobacterium*-mediated transient expression in tobacco (*Nicotiana benthamiana*) leaves was performed as previously described [18]. Images were processed using the laser scanning confocal microscopy (Olympus FV3000, Tokyo, Japan).

### 4.7. Plant Material and Stress Treatment

According to the previous study [45], *B. napus* cv. “Zhongshuang 11” (ZS11) was planted and treated for qRT-PCR analysis. The seeds were sterilized and placed a wet filter paper for germination. Ten days later, seedlings were cultured in hydroponics with modified Hoagland’s nutrient solution (NO_3_^−^, 7.5 mM). After two weeks, they were used for low nitrogen treatment (NO_3_^−^, 0.19 mM) and ABA treatment (100 uM) under the condition of 16 h light, 8h dark and 5000 Lux illumination intensity at 25 °C. Roots and leaves were sampled after 3 h, 12 h, 24 h, 3 d and 7 d nitrogen treatment from high nitrogen level (NO_3_^−^, 7.5 mM) and low nitrogen level (NO_3_^−^, 0.19 mM). Similarly, Roots and leaves were sampled after 3 h, 12 h and 24 h treatment with or without 100 uM ABA. All samples were collected and stored in the −80 °C.

### 4.8. Gene Expression Analysis of SLAC/SLAHs in B. napus

Twelve different tissues and organs based on the RNA-seq data from the *B. napus* ‘Zhongshuang 11′ (BioProject ID PRJNA394926) were downloaded from NCBI-SRA database (https://www.ncbi.nlm.nih.gov/sra/, accessed on 26 January 2021) for further analysis. Total RNA was isolated from the collected tissues using a total RNA extracting kit (Bioteke, Beijing, China) according to the manufacturer’s instructions. Gel electrophoresis was used to determine the quality of RNA samples (Appendix A). Then 1 μg RNA of each sample was used to synthesize cDNA with the PrimeScript RT reagent Kit (Trans Gen, Beijing, China). The specific primers of the *BnSLAC*/*SLAH* genes and the housekeeping gene (*Bnactin7*) were designed using the Primer Premier 5.0 software (Appendix A). The NCBI-Primer BLAST against the rapeseed cDNA database and the melt curves (Appendix A) corresponding primers were examined to verify the specificity. The qRT-PCR assays were performed with three replicates. All of the real-time PCR reactions included 2 μL diluted cDNA, 200 nM of each primer, 2× SYBER GREEN Master Mix (Bioteke, Beijing, China) and sterile water, for a final volume of 20 μL. The following thermal cycle conditions were used with the Applied Biosystems (ABI Quantstudio 5, Foster city, CA, USA): pre-incubation at 94 °C for 2 min, then 45 cycles of 94 °C for 15 s, 60 °C for 15 s and 72 °C for 30 s, and with a default procedure for melt curve. Additionally, the relative expression levels were calculated with the 2^−ΔΔCt^ method. Data was analyzed using the Office 2010 software, and statistical analyses were conducted with SPSS17.0 software using Student’s *t* test at the *p* < 0.05 level of significance.

## 5. Conclusions

In conclusion, 23 full-length *SLAC/SLAH* genes were identified in *B. napus* genomes. The subcellular localizations of BnSLAC/SLAHs provided evidence regarding their functions across the plasma membrane. Moreover, RNA-seq and qRT-PCR analysis showed that *BnSLAH3-2*, *BnSLAH3-3* and *BnSLAH3-4* genes were highly expressed in stamens, pistils, roots, leaves, silique and sepals, and they may be involved nitrate uptake and transport in *B. napus*. The characteristics and analysis of gene structure, physic-chemical properties, homologs phylogeny and experimental data provide a framework for further analysis of the *BnSLAC/SLAH* genes to define their biological functions during stress responses, as well as growth and development in *B. napus*.

## Figures and Tables

**Figure 1 ijms-22-04671-f001:**
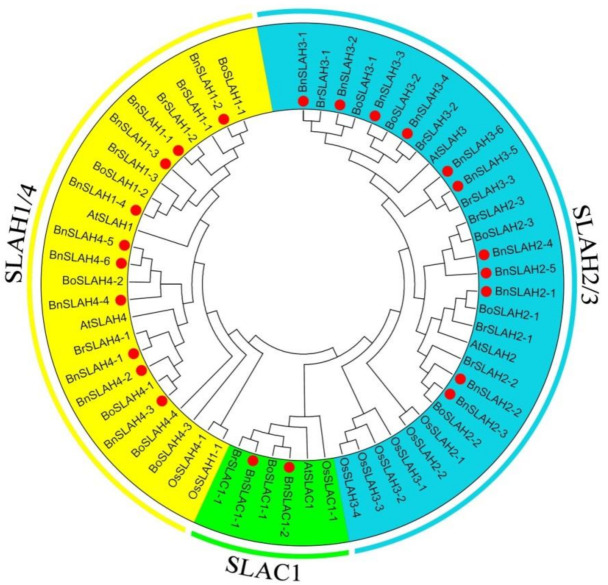
Phylogenetic tree of SLAC/SLAHs among *B. napus*, *B. rapa*, *B. oleracea*, *A. thaliana* and *O. sativa*. The tree was generated using MEGA 6.0 with the Neighbor-Joining method. The proteins clustered into three subgroups. Yellow, blue and green sections indicate the three subfamilies of the SLAC/SLAH proteins. *B. napus* SLAC/SLAHs were marked with red dot.

**Figure 2 ijms-22-04671-f002:**
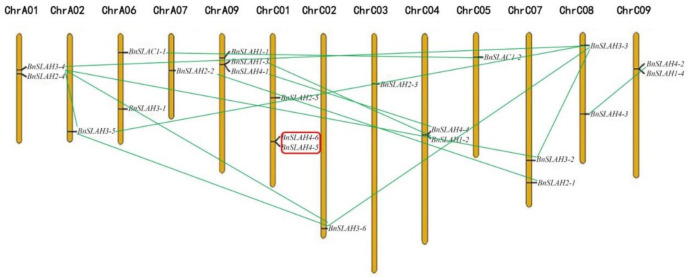
Chromosomal localizations and gene duplication of *BnSLAC*/*SLAH* genes. Tandem duplication genes were circled with red line, and segmental duplication were linked with green lines.

**Figure 3 ijms-22-04671-f003:**
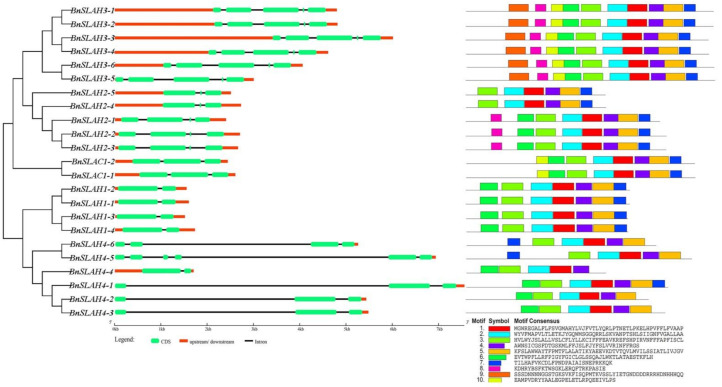
The phylogenetic relationships, gene structures and conserved motifs of SLAC/SLAHs in *B. napus*.

**Figure 4 ijms-22-04671-f004:**
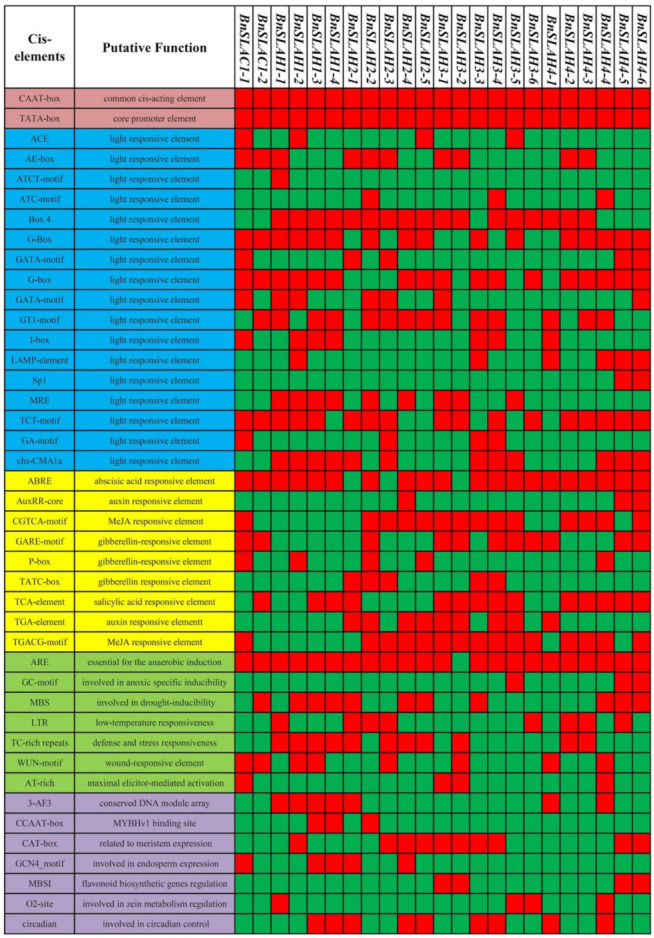
Cis-elements in the 2.0-kb upstream regions of the *BnSLAC*/*SLAH* family genes in *B. napus*. On the left, five different color (brown, blue, yellow, green and purple) represent various types of cis-elements, including core promoter elements, light responsive, phytohormone responsive, abiotic stress responsive and others related to growth. On the right, Red square shows the presence of cis-element while the green indicates the absence of cis-element

**Figure 5 ijms-22-04671-f005:**
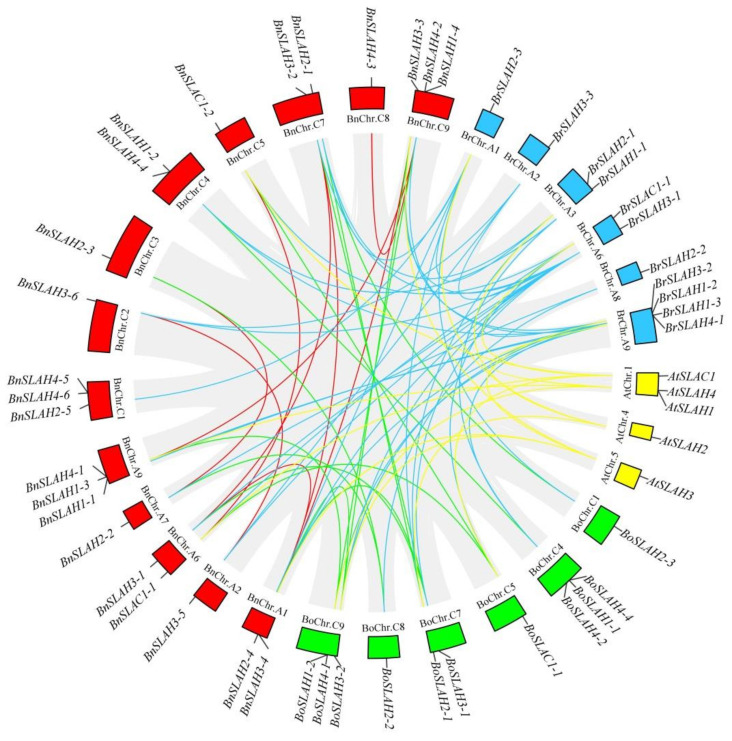
Syntenic relationships of *SLAC/SLAH* genes among *B. napus*, *B. rapa*, *B. oleracea* and *Arabidopsis*. The circular forms of Brassica species and *Arabidopsis* chromosomes are shown in different colors. The approximate positions of the *SLAC/SLAH* genes are marked with short black lines on the circles. Gene pairs with syntenic relationships are joined by the colored lines.

**Figure 6 ijms-22-04671-f006:**
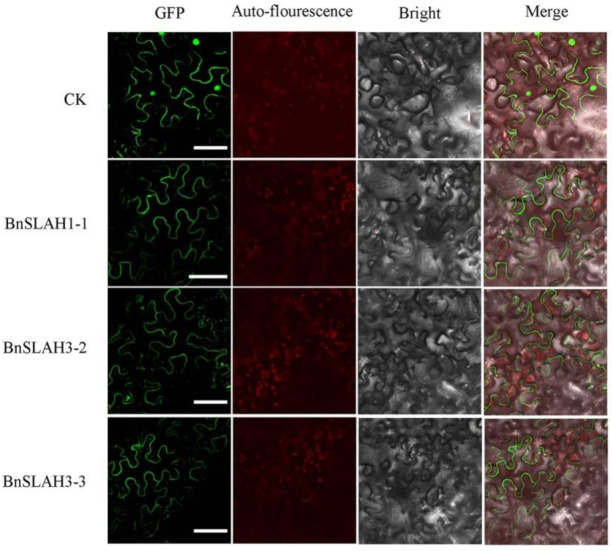
Subcellular localization of BnSLAC/SLAHs in *B. napus*. The selected BnSLAC/SLAH-GFP fusion proteins (BnSLAH1-1-GFP, BnSLAH3-2-GFP and PbrSLAH3-3-GFP) as well as 35S-GFP as the control were independently transiently expressed in tobacco leaves and imaged under a confocal microscope. Bars = 50 μm.

**Figure 7 ijms-22-04671-f007:**
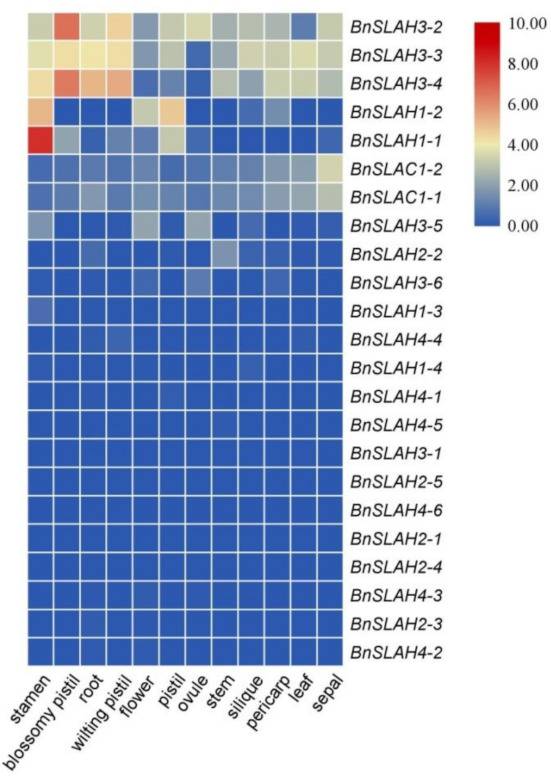
Expression patterns of *BnSLAC/SLAH* family genes in twelve tissues. The expression data were gained from the RNA-seq data and calculated by fragments per kilobase of exon model per million (FPKM) values. The label above the heatmap represents the different tissues of *B. napus* ZS11, the right side of the heatmap represents different *BnSLAC/SLAH* genes. The colour bar represents log_2_ FPKM values.

**Figure 8 ijms-22-04671-f008:**
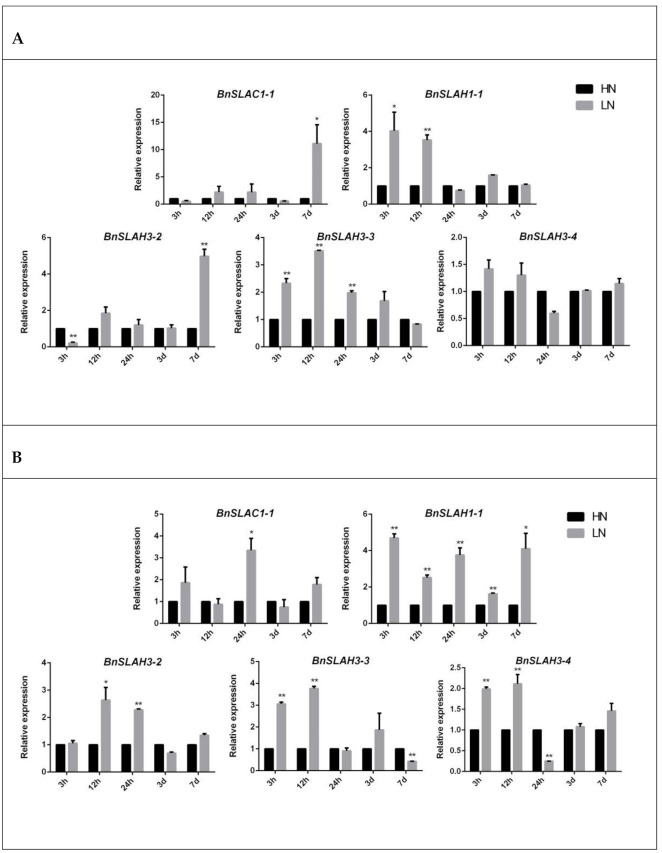
The expression patterns of selected *BnSLAC/SLAH* genes in leaves (**A**) and roots (**B**) after 3 h, 12 h, 24 h, 3 d and 7 d HN (7.5 mM) and LN (0.19 mM) treatment. The asterisk represents statistical significance (Student’s *t*-test, “*” *p* < 0.05, “**” *p* < 0.01) in comparison with control. The data are shown as mean values ± SE (*n* = 3).

**Figure 9 ijms-22-04671-f009:**
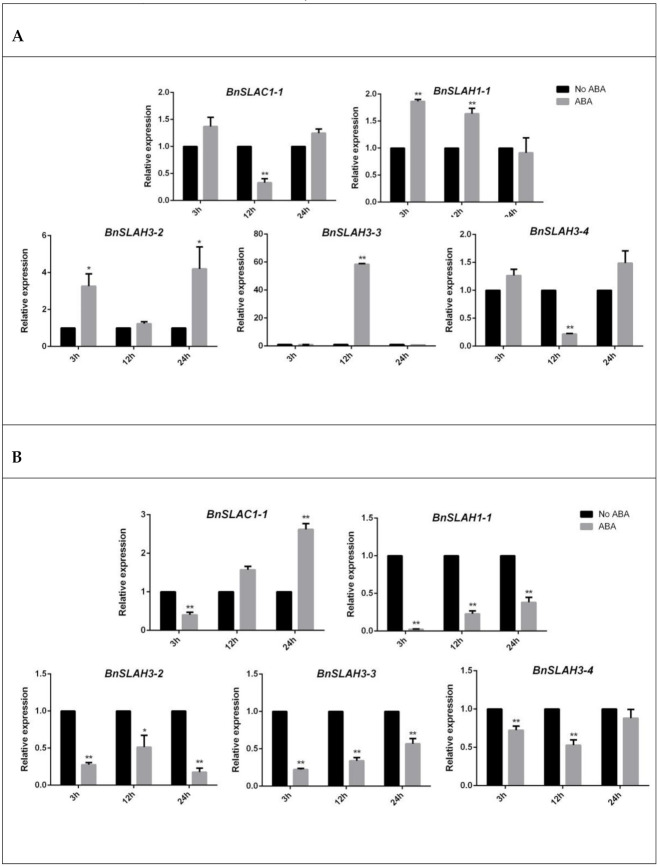
The expression patterns of selected *BnSLAC/SLAH* genes in leaves (**A**) and roots (**B**) after 3 h, 12 h and 24 h ABA (100 uM) treatment. The asterisk represents statistical significance (Student’s *t*-test, “*” *p* < 0.05, “**” *p* < 0.01) in comparison with control. The data are shown as mean values ± SE (*n* = 3).

**Table 1 ijms-22-04671-t001:** Characterization of 23 SLAC/SLAH members identified in *B. napus*.

Gene Features				Protein Features				
Putative Gene Name	Gene ID	Exon No.	Chr. Location	Protein Length (aa)	MW (KDa)	pI	GRAVY	TMD
*BnSLAC1-1*	LOC106445577	3	A06 (5554449..5557045)	588	66.64	9.22	−0.046	8
*BnSLAC1-2*	LOC111206299	3	C05(7993671..7996097)	586	66.18	9.3	0.03	8
*BnSLAH1-1*	LOC106365268	2	A09(7429214..7430811)	387	43.40	9.28	0.5	8
*BnSLAH1-2*	LOC106373608	2	C04(33942406..33943955)	387	43.81	9.29	0.455	8
*BnSLAH1-3*	LOC106373008	2	A09 (9740564..9742068)	381	42.99	9.27	0.566	8
*BnSLAH1-4*	LOC106400098	2	C09 (12333910..12335636)	381	43.04	9.11	0.551	8
*BnSLAH2-1*	LOC106409845	4	C07 (54435207..54437674)	498	56.26	9.3	0.231	10
*BnSLAH2-2*	LOC106359835	4	A07(11635779..11638473)	515	57.78	9.65	0.234	10
*BnSLAH2-3*	LOC106413896	4	C03 (16311421..16314068)	515	57.80	9.54	1.235	10
*BnSLAH2-4*	LOC106452714	3	A01 (12909477..12912194)	359	40.31	9.44	0.626	9
*BnSLAH2-5*	LOC106376145	3	C01 (21092721..21095213)	359	40.36	9.44	0.613	9
*BnSLAH3-1*	LOC106348290	5	A06 (25034448..25039244)	636	72.42	8.34	−0.092	9
*BnSLAH3-2*	LOC106436465	5	C07 (46161649..46166453)	636	72.41	8.39	−0.111	10
*BnSLAH3-3*	LOC106428459	5	C09 (3799924..3805933)	623	71.08	6.88	−0.105	9
*BnSLAH3-4*	LOC106366019	5	A01(11440762..11445373)	623	71.07	7.07	−0.11	9
*BnSLAH3-5*	LOC106415253	5	A02 (32761829..32765883)	640	72.59	8.95	−0.07	9
*BnSLAH3-6*	LOC106378992	5	C02 (66036653..66039643)	636	72.24	8.88	−0.074	9
*BnSLAH4-1*	LOC106417979	3	A09 (9746804..9754357)	476	54.27	9.26	0.308	10
*BnSLAH4-2*	LOC106397706	3	C09 (12312761..12318196)	470	53.58	9.22	0.32	10
*BnSLAH4-3*	LOC106412201	3	C08(29119049..29124524)	470	53.60	9.21	0.324	10
*BnSLAH4-4*	LOC106395319	2	C04 (33847294..33848994)	329	37.57	8.97	0.649	6
*BnSLAH4-5*	LOC106378846	6	C01(36168496..36175425)	533	60.56	6.04	0.19	8
*BnSLAH4-6*	LOC106378847	4	C01(36145965..36151210)	447	51.01	6.43	0.277	8

**Table 2 ijms-22-04671-t002:** The synonymous substitution rates (Ks) and non-synonymous substitution rates (Ka) of the SLAC/SLAHs in *A. thaliana* and *B. napus*.

Gene Name in *A. thaliana*	Gene Name in *B. napus*	Ka	Ks	Ka/Ks
*AtSLAC1*	*BnSLAC1-1*	0.0616	0.4541	0.1356529
*AtSLAC1*	*BnSLAC1-2*	0.0634	0.4292	0.1477167
*AtSLAH1*	*BnSLAH1-1*	0.0867	0.4671	0.1856134
*AtSLAH1*	*BnSLAH1-2*	0.0958	0.4702	0.2037431
*AtSLAH1*	*BnSLAH1-3*	0.0929	0.4315	0.2152955
*AtSLAH1*	*BnSLAH1-4*	0.0955	0.4182	0.2283596
*AtSLAH2*	*BnSLAH2-1*	0.0567	0.3762	0.1507177
*AtSLAH2*	*BnSLAH2-2*	0.0686	0.4021	0.1706043
*AtSLAH2*	*BnSLAH2-3*	0.0706	0.4137	0.1706551
*AtSLAH2*	*BnSLAH2-4*	0.0762	0.4482	0.1700134
*AtSLAH2*	*BnSLAH2-5*	0.0741	0.46	0.161087
*AtSLAH3*	*BnSLAH3-1*	0.0694	0.4353	0.1594303
*AtSLAH3*	*BnSLAH3-2*	0.0686	0.3896	0.176078
*AtSLAH3*	*BnSLAH3-3*	0.0638	0.4419	0.1443766
*AtSLAH3*	*BnSLAH3-4*	0.0635	0.4699	0.1351351
*AtSLAH3*	*BnSLAH3-5*	0.1084	0.4767	0.2273967
*AtSLAH3*	*BnSLAH3-6*	0.1092	0.491	0.2224033
*AtSLAH4*	*BnSLAH4-1*	0.172	0.0677	2.5406204
*AtSLAH4*	*BnSLAH4-2*	0.1757	0.0679	2.5876289
*AtSLAH4*	*BnSLAH4-3*	0.1756	0.068	2.5823529
*AtSLAH4*	*BnSLAH4-4*	0.1595	0.0989	1.6127401
*AtSLAH4*	*BnSLAH4-5*	0.2183	0.1575	1.3860317
*AtSLAH4*	*BnSLAH4-6*	0.2183	0.1575	1.3860317

## Data Availability

The RNA-Seq data used in this study are available in the Sequence Read Archive (SRA) at NCBI (SRA accession: PRJNA394926) repository.

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
