# Peer review of "Identification and Expression Analysis of SLAC/SLAH Gene Family in Brassica napus L."

_ijms, 2021, doi:10.3390/ijms22094671_

Round 1

Reviewer 1 Report

Nan et al. did a comprehensive bioinformatic study of the SLAC/SLAH gene family in Brassica napus. Additionally, they performed expression analyses for five genes and different time points in response to low nitrogen and ABA and localization studies for three. With this, they provide information about the possible evolution and the basis for further studies on the function of these genes in B. napus.

The results are very thorough and will be useful for further research, but the manuscript writing can be improved.

  1. The introduction is very informative. However, I was overwhelmed by all the facts. I suggest that you, rather than giving every detail, re-structure the introduction, giving first a broad statement and then a few examples/details. For example, state that the expression of the genes in Arabidopsis are divers in various tissues, and then give some examples without going too much into detail. In the present form, one cannot distinguish between important and unimportant facts for your study. Some of the details in the introduction can be used for the discussions part.
  2. Please introduce, why anion channels are important for stomatal closure and the function of guard cells in that.
  3. Please make it clearer, that SLAC/SLAHs have a dual function in stomatal closure and in nitrate uptake.
  4. Your last part of the discussions suggest that you wanted to focus on the proteins that might function as nitrate transporters. I did not really get that from your introduction. Please focus on this in your introduction.
  5. Do the not-relative expression of the genes in Figures 8 and 9 confirm the expression of the RNA-Seq analysis in Figure 7? For example, very low/no expression of SLAH1-1 in roots and leaves and higher expression of SLAH3-2 to 3-4 in roots than in leaves.
  6. 379: Do you suggest that BnSLAH3 genes are involved in nitrate uptake from the soil?
  7. Is there a connection between ABA and nitrate starvation? Was the ABA experiment for stomatal closure?
  8. Please provide figures with better resolution, esp. Figure 3 and 5.

Minor comments:

  • Figure 1: Correct OsSALC1-1 and OsSALH; explain red dots in legend
  • Figure 4: Explain coloring of the cis-elements/putative functions
  • Figure 5: Why numbers after two numbers for Bn?
  • Figure 7: silique instead of slique? Why root1? wilting pistil (instead of pitil?)
  • Figure 8 and 9: How many replications were used? In both Figures, root and leaves have a different order of graphs, which is confusing. In Fig. 8 SD is given, in Fig. 9 SE?
  • In Table 1, you could delete the column “Protein domain” and possibly also “Predicted subcellular localization”. In the text, you could differentiate more between the subgroups in the feature descriptions. What is the unit of “Protein length”?
  • Be consistent with plant names: First time write Brassica, then only B.; everytime in italics; also write that you mean Arabidopsis thaliana when you write only Arabidopsis for the first time.
  • Please explain every abbreviation at first use.
  • Check the English language critically, for example wrong use of “including”. l. 364 and 371: compared?
  • 86: poplar?
  • 183-185: substitute “-“ with “to”
  • 204: Please be aware, that you cannot be sure, that the full 2-kb upstream region is the promoter region of each gene
  • 216: Figure S2 is missing. Figure S1 is, what Figure S2 should be, but for Arabidopsis. Is OZS1=SLAC in Fig. S1?
  • 229: Figure 5?
  • 651: BnSLAH1, which one?
  • 279/298: rapeseed, why in italics?
  • 284-288: I think the expression patterns in roots are not sufficiently explained.

Reviewer 2 Report

The study was focused on genome-wide identification and expression analysis of SLAC/SLAH gene family members were performed in B. napus. A total of 23 SLAC/SLAH genes were identified in Brassica napus L. Based on the structural characteristics and phylogenetic analysis of these members, the SLAC/SLAHs could be classified into three main groups. Transcriptome data revealed that BnSLAH3 genes were detected in various tissues of the rapeseed and could be up-regulated by low nitrate treatment in roots. BnSLAC/SLAHs were exclusively localized on the plasma membrane in transient expression of tobacco leaves. These results will increase understanding of the evolution and expression of the SLAC/SLAHs and provide evidence for further research of biological functions of candidates in B. napus.

In my opinion, the results presented in the manuscript are quite interesting, and provide important basis for further research in the topic. However, some improvements are highly recommended:

  • The Introduction is overloaded with content, therefore, it should be presented in more concise form.
  • Detailed numerical data in few tables (e.g. Table 1) should be presented in the Supplementary File.
  • Graphical quality of figures 2, 3 and 5 should be significantly increased. Text inside the figures is almost illegible.
  • Figure 8 – the type of statistical test used should be added in the caption.
  • Student’s t-test should be changed to ANOVA test with post-hoc analysis. All relevant data should be re-calculated using ANOVA test.
  • Page 17, Line 497: The Authors used “2× SYBER GREEN Master Mix (Bioteke, China)”, SYBR Green fluorescent dye during gene expression studies. In this case, it is obligatory to perform Melting Curve Analysis, and results of this examination should be added in the manuscript or Supplementary file (e.g., JPG or TIFF file).
  • I suggest including electropherograms of RNA samples in the Supplementary File.
  • Discussion of the results is superficial and should be thoroughly revised.
  • Conclusions contain too much repetition of the results. They should be re-written, in a more coherent, condensed form. Please, avoid repetition of the results obtained.
  • English grammar and style usage in the manuscript should be checked and corrected by the native speaker, specialist in the field.
  • In addition, I have noticed some spelling errors.

Round 2

Reviewer 1 Report

All comments have been sufficiently answered.